# Erythrocytes as a Source of Exerkines

**DOI:** 10.3390/ijms26199665

**Published:** 2025-10-03

**Authors:** Francesco Misiti, Lavinia Falese, Alice Iannaccone, Pierluigi Diotaiuti

**Affiliations:** 1Human, Social and Health Department, University of Cassino and Lazio Meridionale, V. S. Angelo, Loc. Folcara, 03043 Cassino, Italy; l.falese@unicas.it (L.F.); alice.iannaccone@unicas.it (A.I.); p.diotaiuti@unicas.it (P.D.); 2European University of Technology, EUt+, European Union

**Keywords:** erythrocyte, exerkines, nitric oxide, ATP, extracellular vesicles, lactate, sphingosine-1-phosphate, ROS, microRNA

## Abstract

Exercise activates many metabolic and signaling pathways in skeletal muscle and other tissues and cells, causing numerous systemic beneficial metabolic effects. Traditionally recognized for their principal role in oxygen (O_2_) transport, erythrocytes have emerged as dynamic regulators of vascular homeostasis. Beyond their respiratory function, erythrocytes modulate vascular tone through crosstalk with other cells and tissues, particularly under hypoxia and physical exercise. This regulatory capacity is primarily mediated through the controlled release in the bloodstream of adenosine triphosphate (ATP) and nitric oxide (NO), two potent vasodilators that contribute significantly to matching oxygen supply with tissue metabolic demand. Emerging evidence suggests that many other erythrocyte-released molecules may act as additional factors involved in tissue-erythrocyte crosstalk. This review highlights erythrocytes as active contributors to exercise-induced adaptations through their exocrine signaling.

## 1. Introduction

In recent years, the term exerkines has emerged in exercise physiology and molecular biology to describe the vast array of signaling molecules secreted by skeletal muscle, heart, liver, adipose tissue, and other endocrine-active organs in response to physical activity [1,2]. These bioactive substances include proteins, metabolites, nucleic acids, extracellular vesicles, and gut microbes that play key roles in mediating the systemic benefits of exercise on metabolic health, inflammation, and tissue regeneration [3,4,5]. Exerkines act in autocrine, paracrine, and endocrine fashions, influencing local and distant tissues [6]. For instance, interleukin-6 (IL-6), one of the first identified myokines, increases significantly during exercise and contributes to glucose metabolism and lipid oxidation [7]. In addition, a range of other exerkines are recognized, including IL7 [8], 12,13-diHOME [9], myonectin [10], myostatin [11], METRNL, CSF1, decorin [12,13,14,15], SFRP4 [5], fetuin-A [16], and ANGPTL4 [17], among many others [5]. Exerkines have various health benefits, including promoting angiogenesis, enhancing endothelial cell function, and potentially rescuing cognitive decline [5,18,19]. These molecules have been linked to the regulation of neurodegenerative diseases and have shown promise in the treatment of Alzheimer’s disease, type 2 diabetes [20,21,22], cancer, and Parkinson’s [23]. Different types of exercise elicit varying endocrine responses. For example, high-intensity interval training can lead to a greater increase in certain exerkines compared to lower-intensity exercise [24,25,26]. Understanding exercise’s effects has been broadened, moving beyond the traditional cells. Research now highlights the impact of exercise on other tissues, like the liver, adipose tissue, and brain, and on systemic responses. This shift emphasizes the importance of inter-tissue communication and the role of muscle as an endocrine organ, secreting signaling molecules that affect distant tissues. Erythrocytes undergo significant mechanical and biochemical stress during exercise owing to increased cardiac output, augmented shear forces, and dynamic changes in blood rheology. These conditions stimulate the release of various erythrocyte-derived factors. Among these, adenosine triphosphate (ATP) and nitric oxide (NO) derivatives are particularly noteworthy [27,28,29,30,31,32]. In addition to ATP, erythrocyte membranes possess surface proteins and transporters that may interact with or respond to circulating exerkines, modulating systemic inflammatory responses during and after exercise [33,34]. These interactions may occur through erythrocyte binding, scavenging, or modification of bioactive molecules. Extracellular vesicles (EVs) shed by erythrocytes during physical activity may carry functional microRNAs, proteins, and lipids that contribute to intercellular communication, particularly in vascular and immune contexts [35]. The emerging paradigm posits that erythrocytes are not simply passive oxygen carriers but are active producers and modulators of exerkines (collectively termed erythrokines by the authors) (Table 1), contributing to systemic adaptation during exercise. This novel viewpoint contrasts traditional models and opens new investigative avenues for understanding exercise physiology and adaptation.

## 2. Erythrocyte-Derived Exerkines

Beyond their respiratory function, erythrocytes modulate vascular tone through crosstalk with other cells and tissues, particularly under hypoxia and physical exercise. This review will elucidate the critical role of various molecules released from erythrocytes during exercise. These include ATP, nitric oxide/cyclic guanosine monophosphate, sphingosine-1-phosphate (S1P), lactate, reactive oxygen species (ROS), microRNA, and erythrocyte-derived microvesicles that are released from erythrocytes during physical exercise and participate with other exerkines in the integrative physiological response to exercise, affecting, among others, vascular function, metabolic regulation, and immune modulation. The field of exerkines is rapidly growing to a size that requires specialization, which has been extensively reviewed elsewhere [3,4,5]. This review focuses on erythrocyte-derived factors as exerkines, i.e., erythrokines, that have been confirmed to be released from erythrocytes during exercise in humans. Furthermore, identifying erythrocytes as an endocrine cell has clinical implications, which are highlighted in the Review, such as the central role that erythrocyte plays in organ crosstalk during exercise (Figure 1).

### 2.1. ATP

Current evidence indicates that erythrocytes directly regulate O_2_ supply to human skeletal muscle during dynamic exercise [36,46,47]. This theory aligns with observations made by two research teams that have shown erythrocytes release ATP and NO [27,28,29,30,31,32] when there is a decrease in hemoglobin oxygen saturation [36,48]. As oxyhemoglobin levels drop, NO is released from the S-nitrosohemoglobin compound, facilitating movement to the blood vessel lining. This, in turn, promotes the relaxation of the blood vessels [37,49]. The mechanistic link between Hb deoxygenation and ATP release is thought to involve deoxyHb binding to the N-terminal cytoplasmic domain of band 3 (also known as AE1), the main erythrocyte membrane protein. The resulting increase in membrane-associated ATP may trigger its release [48]. Kirby et al. [46] suggested that the elevation in extracellular ATP observed in blood during exercise originated from erythrocytes. Extracellular ATP binds to P2Y1, P2Y2, and P2Y4 receptors, triggering vasodilators like nitric oxide that relax smooth muscles and improve local blood flow [30,37]. RBC do express P2 receptors, but the high expression of P2X1 and P2X7, along with low levels of several P2Y receptors, appears to be the reverse of scientific findings; research indicates that while P2X1 and P2X7 are present, and P2Y1, P2Y4, P2Y6, P2Y11, and P2Y12 are generally low or absent, P2Y13 is highly expressed, and P2Y1 is also present in mature human RBCs [50]. Activation of P2 receptors stimulates several signaling pathways in mature erythrocytes, such as signaling pathways mediating volume regulation, eicosanoid release, phosphatidylserine exposure, hemolysis, impaired ATP release, and susceptibility or resistance to infection [50]. Consequently, it can improve blood flow to muscle fibers, delivering more oxygen and nutrients and allowing continuous muscle action. Besides hypoxia, ATP release is triggered by mechanical deformation when erythrocytes pass through narrow vessels, involving G protein activation, adenylyl cyclase, and cAMP accumulation [51,52,53,54]. However, the exact ATP release pathway remains unclear [55]. Components of this pathway include the cystic fibrosis transmembrane conductance regulator (CFTR) [56,57] or pannexin-1 as the final conduit for release [58].

### 2.2. Nitric Oxide/Cyclic Guanosine Monophosphate

NO is another critical signaling molecule that causes local vasodilation [38] and exerts inhibitory effects on platelet activation and aggregation, playing a crucial role in vascular homeostasis and anti-thrombotic defense [39]. It is typically formed in vascular endothelial cells upon various stimuli, the most important during exercise likely being shear stress [59]. Studies have shown that erythrocytes carry an active eNOS, which may produce NO under normoxic conditions [60]. The literature has controversially discussed these findings [61,62]. Two pathways have been proposed to mediate the export of NO from the erythrocytes via band 3, either in a metabolon complex with deoxy-hemoglobin promoting its nitrite reductase activity [63] or by band 3-specific trans-nitrosation from S-nitroso-hemoglobin to other NO acceptors outside the cell [49]. During exercise, nitric oxide released from erythrocytes into the bloodstream is crucial in regulating blood flow and oxygen delivery [38,64]. This widening of vessels increases blood flow, ensuring efficient oxygen transport to active tissues. Sur et al. [65] demonstrated that moderate physical exercise is a sufficient stimulator for activating erythrocyte-NOS, directly resulting in improved cell rheological properties. The molecular mechanisms are highly similar to those observed in endothelial cells.

Exercise-induced shear stress results in activation of the PI3 kinase/Akt kinase pathway that, in turn, activates NOS by phosphorylation at Ser^177^ [65]. Activated NOS generates NO in erythrocytes, which is mandatory to improve deformability. It has been shown that erythrocyte deformability is also regulated by PKC [66]. Furthermore, previous studies [40,67] propose that guanosine 3′,5′-cyclic monophosphate (cGMP) released from erythrocytes following soluble guanylate cyclase (sGC) stimulation, such as hypoxia, enters the cardiomyocytes to activate the downstream signaling pathways sGC/cGMP-dependent protein kinase G (PKG), linked to cardio-protective effects [40]. The presence of sGC in mature erythrocytes has been validated [68]. sGC in erythrocytes may be activated by NO via endothelial cells and/or via NO produced by eNOS in erythrocytes [61], suggesting that increasing NO levels during physical activity may stimulate cGMP release from erythrocytes. This finding is significant because physical exercise is essential for treating cardiovascular diseases. Further research is needed to explore the effects of chronic exercise interventions on the molecular erythrocyte-NOS activation pathways discussed.

### 2.3. Sphingosine-1-Phosphate

Sphingosine-1-phosphate (S1P) is a potent bioactive lipid that plays a significant role in various cellular processes, including growth, proliferation, differentiation, migration, and the suppression of apoptosis [41,69,70]. It is found in high-nanomolar concentrations in human plasma, primarily bound to high-density lipoprotein (HDL) and albumin. In recent years, research has increasingly focused on the impact of exercise on S1P metabolism, revealing a critical, dynamic role for erythrocytes in this process [71,72,73]. Erythrocytes are recognized as one of the principal sources of circulating S1P, alongside vascular endothelial cells [74,75,76,77]. S1P promotes deoxygenated hemoglobin (deoxyHb) anchoring to the membrane, enhances the release of membrane-bound glycolytic enzymes to the cytosol, induces glycolysis and thus the production of 2,3-bisphosphoglycerate (2,3-BPG), an eryt. This erythrocyte-specific glycolytic intermediates O_2_ release [78]. These cells possess high sphingosine kinase (SphK) activity, which is the enzyme responsible for phosphorylating sphingosine into S1P, while having low activity of S1P-degrading enzymes [79]. This unique enzymatic profile allows erythrocytes to store substantial quantities of S1P. Unlike platelets, which release S1P primarily upon activation, erythrocytes constitutively release S1P [79,80,81]. The protein responsible for S1P export from erythrocytes is the major facilitator superfamily transporter 2b (Mfsd2b) [76]. Studies using SphK-deficient mice have demonstrated that S1P plasma levels can be restored through erythrocyte transfusion from wild-type mice, underscoring their importance as a circulating S1P [77]. In the resting state, erythrocytes constitute the major source of circulating S1P, maintaining basal plasma levels by continuously exporting S1P through specific transporters. The concentration gradient between erythrocytes and plasma facilitates this process, with high intracellular levels of S1P supporting a steady efflux into the circulation. However, this dynamic shifts during exercise, when the metabolic demands and mechanical forces acting on erythrocytes change significantly. These bidirectional fluxes are modulated by exercise intensity, duration, and the individual’s metabolic status. Acute moderate-intensity exercise, for example, has been shown to elevate plasma levels of S1P significantly, an effect largely attributed to increased erythrocyte synthesis and release [71]. The upregulation of sphingosine kinase activity within erythrocytes, possibly in response to increased intracellular calcium or mechanical shear stress, accelerates the phosphorylation of sphingosine to S1P, fueling this export process [79,81]. Simultaneously, erythrocytes may reabsorb sphingosine or phosphorylated derivatives under specific conditions, such as during recovery or in response to extreme exercise stimuli. This uptake is less well characterized than the export process [80]. Still, it is hypothesized to be mediated by endocytic mechanisms or membrane transporters whose activity may be upregulated during oxidative stress or shifts in pH and ion gradients, common during prolonged or intense exercise [81]. For instance, plasma S1P levels can fall dramatically during ultra-endurance events, while erythrocyte S1P levels do not change, suggesting a temporally regulated reuptake or intracellular accumulation mechanism [82,83]. Such a response might reflect a protective adaptation of erythrocytes, aiming to preserve S1P stores or modulate membrane integrity and deformability under prolonged physiological stress [79]. Beyond simple transport dynamics, S1P plays crucial roles in modulating erythrocyte function and interactions with other cells [70]. It affects erythrocyte deformability, antioxidant capacity, and membrane structure [79], all vital during exercise when red blood cells must navigate constricted capillaries under hypoxic conditions [84]. S1P also mediates crosstalk with endothelial cells, contributing to vascular tone regulation by promoting NO production and stabilizing endothelial barrier function [85,86]. S1P plays a protective role in erythrocytes [87]. The S1P released by erythrocytes during exercise may thus help orchestrate vasodilation, enhance oxygen delivery, and dampen inflammatory responses predisposing to the onset of pathologies, acting as a key signaling mediator beyond its metabolic origin [73]. Importantly, the role of erythrocyte-derived S1P during exercise differs with exercise intensity. Moderate activity primarily enhances erythrocyte S1P production and release, promoting vascular and metabolic homeostasis [82,83]. In contrast, during high-intensity exercise, the contribution of erythrocytes to plasma S1P levels may diminish, with skeletal muscle, vascular endothelium, and possibly hepatocytes contributing more significantly to circulating S1P [73]. The decoupling of erythrocyte S1P export in this context may reflect cellular protective mechanisms, including preserving membrane integrity or reallocating metabolic resources. Nevertheless, high-intensity efforts still elevate plasma sphingosine levels, likely derived from muscle tissue, which can serve as substrates for extracellular S1P synthesis in the plasma or vascular microenvironment [73]. This complex interplay between erythrocytes and S1P metabolism highlights their underappreciated role as active regulators in exercise physiology. The bidirectional trafficking of S1P and its metabolic precursors, influenced by factors such as ceramidase activity, redox state, and transporter expression, underscores the sophisticated adaptability of erythrocytes [70,81]. Rather than being passive participants, erythrocytes are deeply involved in the orchestration of systemic responses to exercise, from local vascular adaptation to systemic metabolic regulation [73,82]. The complex interplay between S1P/S1PRs, cytokines like IL-6, also points to S1P’s role in modulating muscle exerkines [73]. The functional plasticity of erythrocytes in handling S1P during exercise holds promising implications for therapeutic strategies in non-communicable diseases, where S1P signaling and erythrocyte health are often dysregulated [70,85].

### 2.4. Lactate

Lactate, once dismissed as a mere waste product of anaerobic metabolism, has been redefined as a critical signaling molecule, released by erythrocytes during physical exertion, playing a multifaceted role in intercellular communication, energy metabolism, and adaptive physiological responses [44,88]. Gladden [89] provides an extensive overview of lactate metabolism, presenting a paradigm shift from the previous view of lactate as a simple waste product. It highlights how lactate is now recognized as a crucial metabolic intermediate, a mobile fuel for aerobic metabolism, and a potential mediator of redox status within and between cells. Furthermore, lactate debunks long-standing myths regarding lactate’s role in muscle fatigue, acidosis, and ischemic brain injury, suggesting important functions in wound repair, sepsis, and post-ischemic recovery instead. Erythrocytes, devoid of mitochondria and thus relying exclusively on glycolysis for ATP production, have become a primary source of lactate during exercise, particularly under conditions of heightened energy demand or oxygen limitation (hypoxia) [90]. As exercise intensity escalates, the oxygen supply to active skeletal muscles may become insufficient to meet metabolic needs, prompting a shift toward anaerobic glycolysis, wherein glucose is metabolized into pyruvate and subsequently converted into lactate via the enzyme lactate dehydrogenase (LDH). This lactate is then exported from erythrocytes into the systemic circulation mainly via monocarboxylate transporter (MCT1), where it functions not merely as a metabolic intermediate but as a hormone-like signaling molecule that orchestrates a wide array of physiological adaptations [88]. In this capacity, lactate has been redefined as an exerkine, exerting effects across multiple organ systems [90]. Lactate operates as a potent signaling molecule in exercise, activating key cellular pathways such as the hypoxia-inducible factor-1α (HIF-1α) pathway and brain-derived neurotrophic factor (BDNF) release [89,90,91]. Following acute exercise on an ergometer, MCT1 density in erythrocytes was found to be increased [92]. This suggests that strength training can alter how MCT1 responds to exercise, potentially improving lactate influx from plasma to erythrocytes. This could be beneficial during exercise, as lactate is produced as a byproduct of glycolysis by other tissues and released into circulation.

Lactate also binds to the G-protein-coupled receptor 81 (GPR81), expressed in adipose tissue and other tissues, although in lower levels, including the brain [93]; lactate binding to GPR81exerts anti-inflammatory and insulin-sensitizing effects, reducing exercise-induced muscle damage and enhancing glucose uptake in skeletal muscle [94,95]. This mechanism may explain some long-term metabolic benefits of regular physical activity.

Furthermore, lactate has been implicated in immune modulation, suppressing the release of pro-inflammatory cytokines (e.g., TNF-α, IL-6) while promoting the polarization of macrophages toward an anti-inflammatory (M2) phenotype, thereby facilitating tissue repair and recovery post-exercise [96,97]. Recent advancements have uncovered lactate’s role in epigenetic regulation, particularly through a novel post-translational modification termed histone lactylation, wherein lactate-derived lactyl groups are incorporated into histones, altering chromatin structure and modulating gene expression in response to metabolic stress [96,98,99]. This finding has profound implications for understanding how exercise induces long-term adaptations at the genetic level, potentially influencing endurance capacity, mitochondrial biogenesis, and metabolic flexibility. Our hypothesis to consider lactate released from erythrocytes as an exerkine, yet suggested for lactate-derived muscle [90], fundamentally challenges the outdated “lactate threshold” model, which erroneously linked lactate accumulation solely to muscle fatigue and acidosis, instead highlighting its essential role as a master regulator of exercise physiology that integrates metabolic, cardiovascular, immune, and neurological responses to physical activity. Given these diverse and systemic roles, the release of lactate from erythrocytes during exercise represents a crucial adaptive mechanism that enhances performance, promotes recovery, and mediates many of the health benefits associated with regular physical activity.

### 2.5. Extracellular Vesicles (EVs) and microRNA

Due to their abundance and lack of DNA, erythrocytes present an ideal source for extracellular vesicles [42,100] production. Erythrocyte-derived EVs (RBCEVs) are produced both during the maturation of reticulocytes into erythrocytes and as part of the aging process of mature erythrocytes [101]. Erythrocytes lack nuclei and the capacity for new protein or lipid synthesis, so EVs production is one of the limited ways these cells modulate their membrane composition [42,100]. Erythrocytes shed about 20% of their membrane during their lifespan via EVs, releasing factors like calcium influx, oxidative stress, and the loss of cytoskeletal support, which trigger vesicle formation [36,102,103,104]. RBCEVs carry a complex cargo, including proteins enriched in membrane proteins and lipids, similar in composition to the parent erythrocyte and RNAs [42]. RBCEVs involve intracellular NO homeostasis, redox balance, procoagulant effects, and immunomodulation [36,103]. Inspired by the growing body of data implicating EVs in the participation of tissue communication, it has been hypothesized that exercise might stimulate EV secretion and provide a mechanism by which tissues can transfer important signaling molecules from one tissue to another [105,106,107]. Nemkov et al. [108] showed that 30 min of high-intensity cycling may increase RBCEV shedding, potentially due to augmented capillary perfusion that increases erythrocyte shear stress exposure. This release of EVs from erythrocytes is influenced by several exercise-related factors, such as mechanical and oxidative stress in erythrocytes, increased intracellular calcium, and membrane modulation [42,100,102]. EV production allows erythrocyte to modify their membrane composition, potentially improving their deformability, which is crucial for their function during exercise. These data suggest that healthy erythrocytes may release EVs in response to altered vascular hemodynamics, as seen during exercise [108], to support vascular function.

Recent research has highlighted the potential of specific miRNAs to act as exerkines. These exercise-induced circulating miRNAs, termed c-miRNAs, have been proposed to contribute to the multisystemic adaptive effects of physical activity [107]. Over a decade ago, researchers identified erythrocyte-derived miRNAs (miR-125b-5p, miR-451a) as a significant source of miRNAs in the blood [109,110]. It has been described that cell-derived miRNAs can be transported within the circulation attached to Ago2 (argonaute RISC catalytic component 2) and high-density lipoprotein or engulfed in different types of extracellular vesicles (EVs). Studies have shown that Ago2 and EVs are the main carriers for erythrocyte-derived miRNAs, which play a crucial role in RBC-to-cardiovascular cell communication [43]. Alterations in miRNAs bound to erythrocytes during exercise might be targets for future therapeutic applications [43,111].

### 2.6. Reactive Oxygen Species

Erythrocytes are continuously exposed to oxidative challenges due to their intrinsic biology [112]. Each erythrocyte transports oxygen via hemoglobin (Hb), and the autooxidation of Hb, particularly during transit through high oxygen-tension environments, inevitably produces superoxide (O_2_^−^) and other ROS [113,114]. Moreover, erythrocytes can uptake extracellular ROS released by other cells in the blood flow. Accumulated ROS can induce structural changes to the cell membrane, impairing erythrocyte function and generating a hypercoagulable milieu [45]. The ROS-generating potential of erythrocytes is further amplified during physical exercise, when oxygen flux increases, and the rate of redox cycling accelerates [115,116,117,118]. In addition, very recently, it has been shown that oxidative stress from erythrocytes affects muscle oxygenation and performance [119]. Exercise also induces systemic ROS production through mitochondrial respiration in active muscles, xanthine oxidase activation, NADPH oxidase activity, and the catecholamine surge, which can influence erythrocytes through the bloodstream [120,121]. Importantly, while ROS generated during exercise were once viewed strictly as damaging byproducts, they function as redox messengers regulating vascular tone, antioxidant defense, and metabolic signaling [121,122]. In this regard, G6PD-deficient mice show enhanced cardiac function and mitochondrial adaptations following exercise. Metabolomic and proteomic analyses reveal improved energy metabolism and protein turnover in their muscles. These findings challenge assumptions about hemolytic risk during exercise in G6PD deficiency, suggesting possible metabolic advantages [123]. This challenges the notion that oxidative stress is strictly detrimental and adds depth to the discussion on erythrocyte adaptation during exercise.

ROS can be considered “non-protein exerkines” in exercise due to their activity in intercellular communication and modulation of stress-responsive pathways [124]. During exercise, increased circulating ROS challenges all antioxidant erythrocyte defenses [125,126,127,128]. Studies show that acute bouts of exercise can transiently deplete GSH and increase lipid peroxidation markers such as malondialdehyde (MDA) and F_2_-isoprostanes in erythrocyte membranes [127]. An emerging concept is that erythrocytes serve as sensors and transducers of systemic redox signals, including those originating from skeletal muscle and endothelium during exercise [121,122]. This is possible due to their circulation through all tissues and their ability to respond to redox cues. Moderate oxidative stress enhances deformability by modulating spectrin and actin interactions, whereas excessive ROS causes crosslinking and stiffening [115]. Lastly, exercise-induced ROS may indirectly regulate vascular tone by influencing erythrocyte-NO dynamics, contributing to the flow-mediated dilation observed during and after exercise [60,63,129]. All these findings point to a broader role for erythrocytes in exercise physiology, as not only transporters of oxygen but as active redox participants in the exerkine network. Interestingly, the concept of “erythrocrine signaling”, which describes the release of signaling molecules from erythrocytes, aligns well with the role of ROS as exerkines. While erythrocytes do not synthesize or secrete ROS in the traditional secretory sense, their capacity to modulate ROS flux, buffer systemic oxidative stress, and influence vascular responses qualifies them as modulators of redox-based intercellular signaling. This is particularly relevant during exercise, when localized and transient oxidative stress needs to be tightly regulated. The erythrocyte may help “translate” the redox state of working muscles into vascular responses that optimize perfusion and oxygen extraction. This conclusion conflicts with Chatzinikolaou et al. [119], who suggest that erythrocyte oxidative stress contributes to muscle fatigue by reducing erythrocyte efficiency.

Furthermore, the interaction between erythrocyte ROS and other exerkines is worth considering [34,130]. In pathological contexts, such as overtraining, aging, or chronic inflammation, the beneficial signaling role of ROS in erythrocytes may be overwhelmed, leading to oxidative damage [115,116,117,118]. These effects underscore the importance of redox balance in maintaining erythrocyte functionality under stress [131]. Conversely, bioenergetic enhancers [118] and/or nutritional and pharmacological strategies that modulate oxidative stress, such as antioxidant supplementation, dietary polyphenols, or exercise periodization, can preserve or enhance erythrocyte health status [125,132].

## 3. Erythrocyte Receptors for Exerkines

While much of the focus in exerkine biology has been on their effects on target organs such as the brain and liver, a less explored but intriguing possibility is the interaction of exerkines with erythrocytes (Table 2). Erythrocytes, in addition to possessing receptors for ATP and nitric oxide, previously discussed, recent evidence indicates that they may function as receptors for exerkines produced by other organs [34], like myostatin (MSTN) [11], interleukin-8 (IL-8), or MCP-1. Recent studies show that myostatin (MSTN) and its receptors are present on the membrane of erythrocytes. However, the levels of these receptors (notably TGF-β RI) can decrease in the absence of myostatin and after exhaustive exercise. Myostatin knockout (removal of MSTN activity) improves erythrocyte antioxidant capacity by accelerating the pentose phosphate pathway, helping red blood cells counteract oxidative stress after intense exercise [133]. The presence of DARC in erythrocytes is particularly relevant. The quantification of DARC (Duffy Antigen/Receptor for Chemokines) on erythrocytes is typically estimated at approximately 1000 to 2000 antigen sites per red blood cell in DARC-positive individuals. However, this number can vary slightly depending on methodology and population studied [134]. DARC binds IL-8, as well as MCP-1 [34,130]. Erythrocyte structure was notably affected in the presence of IL-8, where the morphological changes resembled those typically seen in eryptosis (programmed cell death) [135]. It has been suggested that erythrocytes delay the clearance of bound chemokines related to exercise, prolonging their effects [130].

Nevertheless, erythrocyte-derived microparticles express DARC [136]. Emerging evidence suggests that intact erythrocytes and erythrocyte-derived microparticles could act as vesicles transferring DARC-bound chemokine ligands to sites of vascular damage. Moreover, erythrocytes from healthy individuals express TLR9 on the surface of human red blood cells, with an average surface expression of approximately 8.81% in healthy individuals [137,138] and can bind released mtDNA and CpG-rich DNA after acute high-intensity exercise [139]. However, the literature is conflicting regarding whether mtDNA is preferentially carried in plasma or bound to erythrocytes [137,140,141]. Additionally, although there is no evidence of direct interaction with erythrocytes, adiponectin, an exerkine released from adipose tissue [5], is known to affect erythrocyte rheology. Thus, erythrocytes are suggested to function as a “rheostat” of the signaling pathways triggered by exercise by quenching exerkines.

## 4. Current Uncertainties and Research Challenges

Substantial evidence supports the release and role of various erythrocyte-derived exerkines in response to physical activity. ATP and NO are recognized as critical mediators of vascular function and oxygen supply, with many studies affirming their involvement in vasodilation and enhancing blood flow during exercise. The processes through which ATP is released due to hypoxia and mechanical stress and the cycling of NO via S-nitrosohemoglobin are well-documented through molecular, physiological, and receptor studies [30,36]. S1P has also been established as a significant lipid signaling molecule derived from erythrocytes, affecting cell deformability, glycolytic activity, and endothelial balance [78], particularly during moderate-intensity exercise [82,83]. Lactate has transitioned from being seen merely as a metabolic byproduct to being recognized as an influential exerkine that regulates immune, metabolic, and epigenetic responses [90]. This shift is backed by strong evidence regarding monocarboxylate transporter (MCT1) activity and lactate signaling pathways [92]. However, there are ongoing uncertainties and discussions about the contributions of erythrocytes compared to other tissues in the circulation of S1P, especially during high-intensity exercise. While erythrocytes consistently release S1P, the extent of their contribution versus that of the endothelium and muscle is still debated [82,83]. ROS produced by erythrocytes during exercise is controversial; research reveals a division between beneficial stress and detrimental damage. Moderate levels of ROS can enhance erythrocyte deformability, but excessive amounts are associated with oxidative damage and reduced oxygen transport, leaving the balance and individual differences insufficiently clarified [119,120]. Similarly, the effects of lactate accumulation are under discussion. While it shows anti-inflammatory and epigenetic advantages, some studies still associate elevated lactate levels with adverse effects like muscle fatigue and insulin resistance [90]. The role of extracellular vesicles (EVs) and miRNA released from erythrocytes in systemic adaptations, compared to other cellular sources, remains an active area of research. There are significant experimental gaps in several domains [43,111]. Although their release mechanisms are understood for ATP and NO, measuring their flux across different exercise types is challenging and underexplored. The signaling pathways interacting with exerkines, such as the interaction between erythrocytes and myostatin or chemokine binding via DARC receptors, are acknowledged but have not been thoroughly characterized [34,130]. The physiological significance and regulatory mechanisms of exosome transfer through EVs are still speculative. There are open questions regarding EV composition, functional cargo delivery, and how individual variability in EV profiles may differ after exercise. For erythrocyte-derived miRNA, while the carriers and targets are identified, there is a lack of longitudinal studies tracking adaptation throughout training cycles, and issues with technical biases in miRNA quantification persist [107,108]. Moreover, understanding ROS dynamics and their metabolic effects in pathological conditions, such as aging, metabolic disorders, and overtraining, needs further investigation across animal models and various human populations.

## 5. Conclusions and Future Perspectives

In conclusion, the concept of erythrokines challenges the conventional view of erythrocytes as passive gas transporters. Although much remains to be done, it redefines them as active mediators of exercise-induced systemic adaptation and key players in athletes’ training and recovery. Addressing the above underscored limitations calls for multi-modal experimental designs, improved biomarker tools, and broader population studies. Consensus is needed around standardized methodological approaches for exerkine quantification, and cross-tissue comparative studies are essential to clarify relative contributions and therapeutic relevance. These knowledge gaps open avenues for future research targeting the unexplored regulatory network of erythrocyte-derived exerkines, their interactions with other exercise-responsive tissues, and implications for health, disease, and athletic performance.

## Figures and Tables

**Figure 1 ijms-26-09665-f001:**
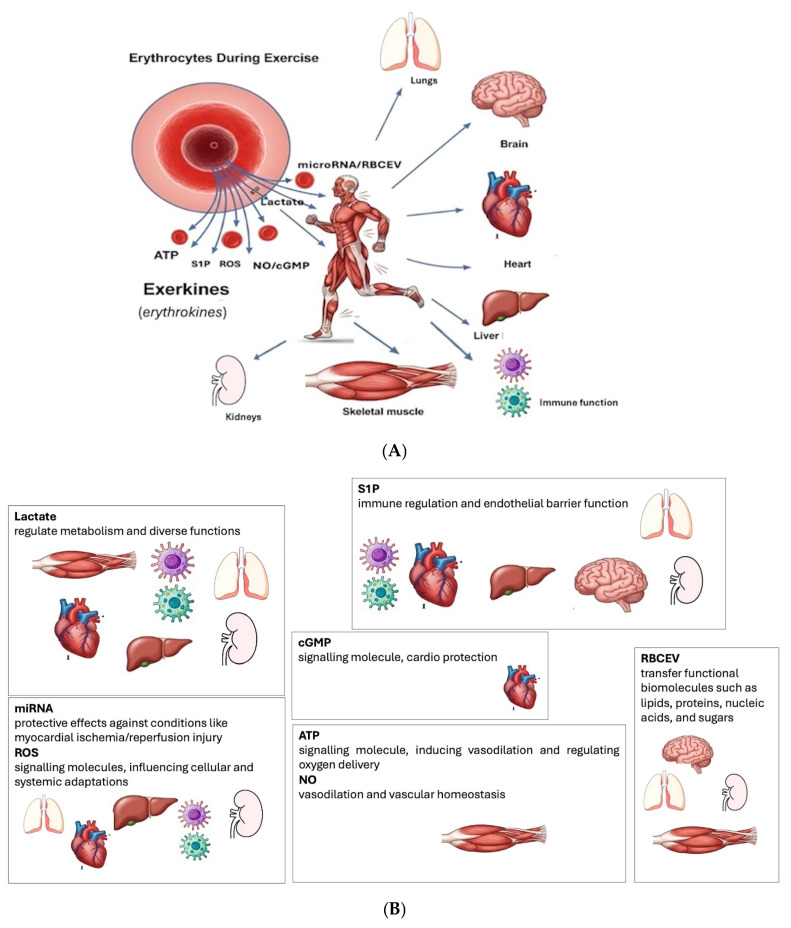
The systemic effects of exerkines released from erythrocytes (erythrokines): (**A**) Key Tissue/Organ affected by ATP (adenosine triphosphate), S1P (sphingosine-1-phosphate), ROS (reactive oxygen species), NO (nitric oxide), cGMP (guanosine 3′,5′-cyclic monophosphate), RBCEV (vesicles derived from erythrocytes); (**B**) Main effects elicited by exerkines released from erythrocytes (erythrokines).

**Table 1 ijms-26-09665-t001:** Exerkines released from erythrocytes.

Exerkines	Molecule Type	Key Tissues Affected
ATP	Organic Molecule	blood vessels (microcirculation) [36,37]
Nitric Oxide (NO)	Inorganic Molecule	smooth muscle cells; platelets [38,39]
Cyclic guanosine monophosphate (cGMP)	Organic Molecule	heart [40]
Sphingosine-1-phosphate (S1P)	Organic Molecule	vascular tissues; lymphatic system; heart; liver; lungs; CNS; kidney [41]
Microparticles or vesicles (RBCEV)	Vesicles	endothelial cells; immune system; lungs; muscle; brain [42]
MicroRNAs (miRNAs)	RNA	bone marrow; lungs; liver; spleen; kidney [43]
Lactate	Organic Molecule	muscle; heart; kidney; liver; brain; adipose tissue; immune system [44]
Reactive oxygen species (ROS)	Inorganic Molecule	microvascular endothelium; platelets [45]

**Table 2 ijms-26-09665-t002:** Receptors for exerkines in erythrocytes.

Exerkines	Receptor
ATP	P2X7/P2Y1 purinergic receptors [50]
Nitric oxide (NO)	Soluble guanylate cyclase (sGC) [68]
Lactate	Monocarboxylate transporter (MCT1) [92]
CXCL8 (IL-8)	Duffy antigen receptor for chemokines (DARC) [34,136]
CCL2 (MCP-1)	Duffy antigen receptor for chemokines (DARC) [34,136]
Myostatin (MSTN)	TGF-β RI [133]
Mitochondrial DNA (mtDNA)	Toll-like receptor 9 (TLR9) [137,138]
microRNA (c-miRNA)	Argonaute RISC catalytic component 2 (Ago2) [43]

## Data Availability

No new data were created or analyzed in this study. Data sharing is not applicable to this article.

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
