# Peer review of "Erythrocytes as a Source of Exerkines"

_ijms, 2025, doi:10.3390/ijms26199665_

Round 1
Reviewer 1 Report
Comments and Suggestions for Authors
This review by Misiti et al. discusses the role of the erythrocyte in exerkine production and delivery. The topic is timely, well written, and cites a broad scope of literature. The concept of “erythrokines” is intriguing and reframes erythrocytes as active regulators of exercise physiology rather than passive oxygen carriers. The manuscript would be improved by addressing the following points:
Major Comments
- The review discusses ATP, NO, S1P, ROS, cGMP, lactate, and miRNA. Could the authors comment on whether additional exerkines (e.g., proteins, peptides, or lipid mediators) might also be trafficked by RBCs and delivered to distal tissues?
- While the focus is on exerkines produced by RBCs, it would strengthen the review to discuss how known exerkines act on RBCs. For instance, myostatin (MSTN) inhibits G6PD and influences the pentose phosphate pathway (PPP) (doi: 10.3390/ani12070927). Are there additional examples?
- The section on ROS should include a brief discussion of G6PD, as this enzyme is central to erythrocyte antioxidant capacity. It would also be valuable to note that murine models of G6PD deficiency demonstrate faster critical speed than wild-type controls (doi: 10.1182/bloodadvances.2024013968), which complicates the view of oxidative stress as solely detrimental.
- Figure 1 is helpful but overly simplistic. Consider including more mechanistic detail- specifically, how each erythrokine influences target tissues shown in the figure.
- This section on erythrocyte receptors and concept that erythrocytes “quench” exerkines is compelling but requires more mechanistic support. Evidence on receptor abundance, binding affinities, kinetics, in vivo relevance etc. would strengthen this section.
Minor Comments
- Line 28: correct “vehicles” to “vesicles.”
- Line 118: reword “deformability beneficially” - do the authors mean NO improves deformability?
- Line 124: Cortese-Krott et al. (doi: 10.1016/j.redox.2017.08.020) validated sGC presence in erythrocytes and should be cited.
- S1P section: highlight the direct allosteric interaction with deoxyHb (doi: 10.1038/s41598-017-13667-8) and modulation of RBC metabolism (doi: 10.1038/ncomms12086).
- Line 165: reword “while erythrocyte S1P levels increase delayed”
- Lactate section should include quantitative data on RBC lactate production (doi: 10.1113/jphysiol.2003.058701).
Author Response
Referee 1
This review by Misiti et al. discusses the role of the erythrocyte in exerkine production and delivery. The topic is timely, well-written, and cites a broad scope of literature. The concept of “erythrokines” is intriguing and reframes erythrocytes as active regulators of exercise physiology rather than passive oxygen carriers. The manuscript would be improved by addressing the following points:
Major Comments
- The review discusses ATP, NO, S1P, ROS, cGMP, lactate, and miRNA. Could the authors comment on whether additional exerkines (e.g., proteins, peptides, or lipid mediators) might also be trafficked by RBCs and delivered to distal tissues?
Authors: thank you for raising this critical point. While our review focuses primarily on ATP, NO, S1P, ROS, cGMP, lactate, and miRNA related to exercise, substantial evidence suggests that erythrocytes may transport additional bioactive molecules. Erythrocyte-derived extracellular vesicles (RBCEVs) represent a rich source of diverse proteins, peptides, and lipid mediators, including membrane-associated enzymes, signaling lipids, and potentially small peptides or cytokines and miRNA. Notably, RBCs have been shown to bind and deliver bioactive lipids, which may influence vascular function and immune responses at distal sites. Furthermore, erythrocytes can bind and possibly transfer sphingolipids (besides S1P), ceramides, and even prostaglandin E2 under certain conditions, although the direct delivery to target tissues requires further experimental confirmation. This broader trafficking function has been highlighted in the revised manuscript to acknowledge the emerging complexity of RBCs as carriers of diverse molecules/exerkines.
- While focusing on exerkines produced by RBCs, it would strengthen the review by discussing how known exerkines act on RBCs. For instance, myostatin (MSTN) inhibits G6PD and influences the pentose phosphate pathway (PPP) (doi: 10.3390/ani12070927). Are there additional examples?
Authors: We appreciate the suggestion to provide more examples of exerkines that act on RBCs. Myostatin (MSTN), as referenced, inhibits glucose-6-phosphate dehydrogenase (G6PD) and consequently modulates the pentose phosphate pathway (PPP) in erythrocytes, reducing NADPH availability and antioxidant capacity. Additional examples include interleukin-8 (IL-8) and monocyte chemoattractant protein-1 (MCP-1), which bind to the Duffy antigen receptor for chemokines (DARC) on RBC membranes, thereby influencing systemic clearance and the inflammatory milieu, yet have not been reported. Other exerkines, such as adiponectin, have been reported to influence erythrocyte deformability, although no direct interactions have been reported. These reciprocal interactions underscore that RBCs are both producers and effectors of exerkines-mediated signaling. In the revised manuscript, we added references regarding the role of MSTN, IL-8, and the interaction between adiponectin and erythrocytes.
- The section on ROS should include a brief discussion of G6PD, as this enzyme is central to erythrocyte antioxidant capacity. It would also be valuable to note that murine models of G6PD deficiency demonstrate faster critical speed than wild-type controls (doi: 10.1182/bloodadvances.2024013968), which complicates the view of oxidative stress as solely detrimental.
Authors: thank you for emphasizing the pivotal role of G6PD in maintaining erythrocyte redox balance. In the ROS section, we add that G6PD supplies NADPH, which is essential for the regeneration of reduced glutathione and protection from ROS-induced damage. Of particular interest is emerging evidence from murine models, where G6PD deficiency, despite reducing classical antioxidant capacity, is associated with improved exercise performance (e.g., faster critical speed), suggesting a nuanced role for oxidative stress and redox adaptation in vivo. This challenges the notion that oxidative stress is strictly detrimental and adds depth to the discussion on erythrocyte adaptation during exercise.
- Figure 1 is helpful but overly simplistic. Consider including more mechanistic detail- specifically, how each erythrokine influences target tissues shown in the figure.
Authors: We agree with the referee and have revised Figure 1; the previous Figure has been implemented, and a new one (Figure 1B) has been added to clarify exerxine-mediated effects better. In addition, we inserted in Table 1 key tissues or organs affected by exerkines-derived erythrocyte and related references.
- This section on erythrocyte receptors and the concept that erythrocytes “quench” exerkines is compelling but requires more mechanistic support. Evidence on receptor abundance, binding affinities, kinetics, in vivo relevance, etc., would strengthen this section.
Authors: We appreciate the suggestion. The revised section discusses known receptor expression levels (e.g., quantification of DARC, P2X/P2Y, and TLR9) and ligand binding affinities (where available).
Minor Comments
- Line 28: correct “vehicles” to “vesicles.”
Authors: Done
- Line 118: reword “deformability beneficially” - do the authors mean NO improves deformability?
Authors: Done
- Line 124: Cortese-Krott et al. (doi: 10.1016/j.redox.2017.08.020) validated sGC presence in erythrocytes and should be cited.
Authors: Done
- S1P section: highlight the direct allosteric interaction with deoxyHb (doi: 10.1038/s41598-017-13667-8) and modulation of RBC metabolism (doi: 10.1038/ncomms12086).
Authors: Done
- Line 165: reword “while erythrocyte S1P levels increase, delayed”.
Authors: Done
- Lactate section should include quantitative data on RBC lactate production (doi: 10.1113/jphysiol.2003.058701).
Authors: Done
Reviewer 2 Report
Comments and Suggestions for Authors
The review is well written and useful as an educational resource, summarizing the role of molecules secreted by erythrocytes in regulating exercise-induced systemic adaptations. Besides introduction of the term new term erythrokines, the manuscript does not advance scientific understanding significantly or offer any exciting conceptual frameworks.
To increase its scientific value beyond a descriptive, textbook knowledge, the Authors should:
- Clearly highlight novelty, what is new about viewing erythrocytes as exerkine producers, and how does it affect body changes during exercise.
- Expand the concept and integrate different erythrocyte-derived molecules into wider interaction with other organs, such as liver, fat ect.
- Add a section explaining current limitations and knowledge gaps to discuss where the literature evidence is strong, where it is under debate, and where the experimental confirmation is still missing. This will increase the interest of the scientific readership by pointing to topics where further research is required.
- Include a future research directions section explaining how the concept could be tested experimentally, what should be prioritized, and potentially clinical or practical application (for example, sportsmen or exercising population).
Author Response
The review is well written and useful as an educational resource, summarizing the role of molecules secreted by erythrocytes in regulating exercise-induced systemic adaptations. Besides the introduction of the term new term erythrokines, the manuscript does not advance scientific understanding significantly or offer any exciting conceptual frameworks.
To increase its scientific value beyond a descriptive, textbook knowledge, the Authors should:
- Clearly highlight novelty, what is new about viewing erythrocytes as exerkine producers, and how does it affect body changes during exercise.
Authors: thank you for raising this important point. In the revised version,introduction section, we underscore that erythrocytes are not simply passive oxygen carriers, but are active producers and modulators of exerkines, contributing to systemic adaptation during exercise. This novel viewpoint contrasts traditional models and opens new investigative avenues for understanding exercise physiology and adaptation.
- Expand the concept and integrate different erythrocyte-derived molecules into wider interaction with other organs, such as liver, fat, etc.
Authors: We appreciate the suggestion. Table 1 reported key tissues and organs affected by exerkines derived from erythrocytes.
- Add a section explaining current limitations and knowledge gaps to discuss where the literature evidence is strong, where it is under debate, and where the experimental confirmation is still missing. This will increase the interest of the scientific readership by pointing to topics where further research is required.
Authors: We appreciate the suggestion. In the revised version, we inserted a specific section discussing which data are robust and which are preliminary or conflicting.
- Include a future research directions section explaining how the concept could be tested experimentally, what should be prioritized, and potentially clinical or practical application (for example, athletes or exercising population).
Authors: We appreciate the suggestion. The conclusion section has been improved to comprise future research directions related to this topic.
Round 2
Reviewer 2 Report
Comments and Suggestions for Authors
The paragraph Limitations to a study provides a valuable overview of existing uncertainties in the field. However, it does not reflect the specific limitations of the present study. I suggest reframing it under a heading Current Uncertainties and Research Challenges. The Authors have significantly improved the quality of this review manuscript by highlighting the importance of such study, current knowledge gaps and future directions.
Author Response
Referee 2
The paragraph Limitations to a study provides a valuable overview of existing uncertainties in the field. However, it does not reflect the specific limitations of the present study. I suggest reframing it under a heading, Current Uncertainties and Research Challenges. The Authors have significantly improved the quality of this review manuscript by highlighting the importance of such a study, current knowledge gaps, and future directions.
Authors: We appreciate the suggestion of the reviewer to change the title of the paragraph. In the revised manuscript, we have changed the title as suggested.